# Enhanced third-order optical nonlinearity in a dipolar carbene-metal-amide material with two-photon excited delayed fluorescence

Ikechukwu D. Nwosu[1], Lujo Matasović[2], Tárcius N. Ramos [3], Nguyen Le Phuoc[4], Giacomo Londi [5], Alexander J. Gillett [2], Daniel T. W. Toolan [6], Charles T. Smith [7], George F. S. Whitehead [1], Mireille Blanchard-Desce[8], Jonathan Daniel [8] ✉, Mikko Linnolahti [4] ✉, Yoann Olivier [9] ✉ & Alexander S. Romanov [1] ✉

Advanced photonic materials showing two-photon absorption (2PA) have been widely explored to develop three-dimensional imaging, micro and nanofabrication, all-optical switching, lithography on a nanoscale and many other enabling technologies. These all require nonlinear absorption chromophores with intrinsic 2PA cross-sections and long-term photo- and thermal stability. Here, we disclose the very first example of the dipolar carbene-metal-amide (CMA) material showing a enhanced 2PA cross-section up to 105 GM. Overall molecular design considerations such as extended π-conjugation (to increase polarizability), minimizing the singlet-triplet energy gap ($\Delta E_{ST}$), and using heavy metal atoms are the first design principles to obtain bright one- and two-photon excited thermally activated delayed fluorescence (TADF) material, showing one of the highest radiative rate of $2.18 \cdot 10^6 \, s^{-1}$ across CMA materials. Bright red CMA 2P-TADF material shows excellent photostability ($LT_{50} = 3$ h) to 20 mW femtosecond pulsed laser excitation at 1000 nm, encouraging further CMA exploration for future applications in advanced photonic technologies requiring third-order nonlinear optical properties.

Photonic materials with high third-order nonlinear optical (NLO) response such as two-photon absorption (2PA) and two-photon excited fluorescence (2PEF) are highly sought after for a variety of advanced optical technologies in data storage[1], multi-photon microscopy[2], imaging[3], photodynamic therapy[4], two-photon nanolithography[5], all-optical switching[6], and optical power limiting[7]. Unlike conventional one photon absorption (1PA, Fig. 1a) where a single photon possesses enough energy to enable the $S_0 \rightarrow S_1$ electronic transition, 2PA relies on the combined energy of two photons that

must hit the molecule simultaneously within few femtoseconds according to the Heisenberg's uncertainty principle under the single intermediate state's approximation (Fig. 1a, where the photon energy equals or is slightly above one half of the energy gap)[5]. The probability of the optical transition pumped by 2PA depends on the square of the light intensity and is characterized by the 2PA cross-section ($\sigma_2$) measured in Göppert-Mayer units (GM, 1 GM = $10^{-50}$ cm$^4$ s photon$^{-1}$). Conventional 2 PEF materials are limited only by singlet harvesting mechanism, while two-photon excited

[1]I.D. Nwosu, G.F.S. Whitehead, A.S. Romanov, Department of Chemistry, University of Manchester, Manchester, UK. [2]L. Matasović, A.J. Gillett, Cavendish Laboratory, Department of Physics, University of Cambridge, Cambridge, UK. [3]T.N.Ramos, Theoretical Chemistry Laboratory, Namur Institute of Structured Matter, Université de Namur Rue de Bruxelles, Namur, Belgium. [4]N.L. Phuoc, M. Linnolahti, Department of Chemistry and Sustainable Technology, University of Eastern Finland, Joensuu, Finland. [5]G. Londi, Department of Chemistry and Industrial Chemistry, University of Pisa, Pisa, Italy. [6]D.T.W. Toolan, Department of Materials, University of Manchester, Manchester, UK. [7]C.T. Smith, Department of Physics and Astronomy and the Photon Science Institute, University of Manchester, Manchester, UK. [8]J. Daniel, M. Blanchard-Desce, Institut des Sciences Moléculaires, University of Bordeaux, Centre National de la Recherche Scientifique, Institut Polytechnique de Bordeaux, Talence, France. [9]Y. Olivier, Laboratory for Computational Modeling of Functional Materials Namur Institute of Structured Matter, Université de Namur Rue de Bruxelles, Namur, Belgium. ✉e-mail: jonathan.daniel@u-bordeaux.fr; mikko.linnolahti@uef.fi; yoann.olivier@unamur.be; alexander.romanov@manchester.ac.uk

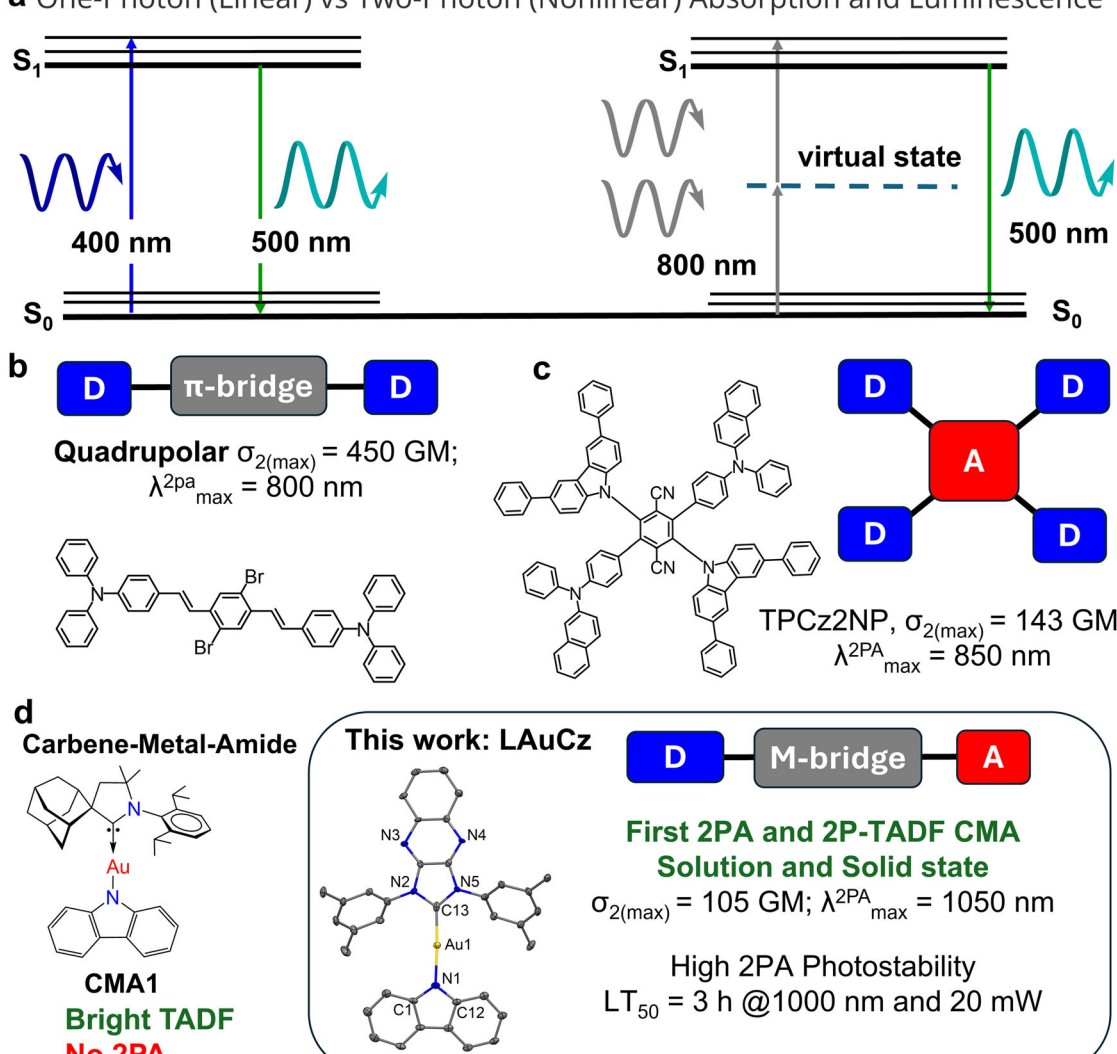

**a** One-Photon (Linear) vs Two-Photon (Nonlinear) Absorption and Luminescence

**Fig. 1 | The molecular design of different two-photon absorption (2 PA) materials, including organic TADF molecules, with their maximum 2PA cross-sections. a** Schematic diagram for one- and two-photon absorption and emission processes; **b, c** Molecular structures of the reported 2P-TADF; **d** molecular structures of carbene-metal-amide (CMA) materials.

thermally activated delayed fluorescence (2P-TADF) materials harvest all singlet and triplet excitons offering a unique advantage and high 2PA brightness ($\sigma_2 \times$PLQY, photoluminescence quantum yield) required for practical applications. This newly emerged interest in such multifunctional materials aims to develop advanced 2P-TADF materials with superior performance[8].

Molecular design strategies towards brighter 2PA chromophores rely on various combinations of the Donor (D), Acceptor (A) moieties and π-conjugated bridge to form a dipolar (D-π-A, V-shaped D-π-A-π-D or A-π-D-π-A), quadrupolar (linear D-π-A-π-D or A-π-D-π-A), octupolar or multipolar materials which facilitates the charge transfer (CT) process[9]. Multiple studies have revealed that quadrupolar and octupolar molecules exhibit much brighter 2PA compared with their dipolar analogues[10]. Such 2PA $\sigma_2$–enhancement has been attributed to much larger polarizabilities for the intramolecular CT state and coherent coupling between branches of the chromophore under the Frenkel exciton model[9]. Based on this design approach, Xu and co-workers[11] synthesized a sky-blue emitting bis(styryl) benzene quadrupolar chromophore (Fig. 1b) with heavy bromine atoms to enhance spin-orbit coupling for efficient triplet harvesting. This molecule exhibited a high 2PA cross-section value of 450 GM at 800 nm, attributed to large changes in the quadrupole moment upon excitation, and sufficient π-

conjugation giving rise to CT excited states. Adachi et al.[8] designed bright multipolar 2P-TADF emitters where molecule TPCz2NP (Fig. 1c) with extended π-conjugation demonstrated large 2PA cross-section up to 143 GM at 850 nm and explaining 2P-TADF due to efficient delayed fluorescence from a hybrid CT/locally-excited (LE) state. Organometallic 2PA materials are well-represented by various alkynyl and aryl complexes of gold(I), ruthenium(II), and platinum(II) which demonstrated particularly strong 2PA cross-section on par with organic materials up to $10^4$ GM for octupolar and multipolar chromophores[12]. In particular, 4d- and 5d-transition metal complexes are the most promising 2PA and 2PEF candidates due to their electron-rich nature and greater polarizability compared with 3d-metal complexes[10]. Unfortunately, the majority of the metal complexes are limited by 2 PA fluorescence or long-lived phosphorescence with no examples of the 2P-TADF materials to date[13].

A vast majority of 2PA measurements are reported in fluid media, while much less works consider solid-state 2PA measurements due to laser-induced damage of the chromophore or the surrounding polymer matrix[14]. The solid state photostability is a natural limiting factor for a number of advanced photonic technologies such as targeted drug delivery[4], two-photon lithography[5] or all optical switching[6] to enable fast switching speeds for telecommunication and computing[15], and enhanced efficiency of

photovoltaic cells[16]. Dipolar molecules usually exhibit lower 2PA cross-sections (0.1–50 GM) compared to quadrupolar molecules (up to $10^4$ GM)[17]. However, molecular design of the robust dipolar (D-π-A) 2PA chromophores is much more feasible compared to multipolar materials. For instance, Hudson[18] and others[19] demonstrated dipolar organic TADF materials with up to 400 GM 2P TADF in toluene solution which are prone to photodegradation within several hours. Therefore, bright dipolar 2PA chromophores with high thermal- and photostability towards pulsed femtosecond laser excitation are one of the most sought-after 2PA materials. At the best of our knowledge, examples of small dipolar molecules exhibiting both 2PA and TADF in the solid state are not reported in the literature neither for organic nor organometallic systems.

Carbene-metal-amides (CMA, M = Cu, Ag or Au) emerged as bright TADF materials enabling energy-efficient organic light emitting diodes (OLEDs)[19]. One of the first successful CMA materials (CMA1, Fig. 1d) was demonstrated based on a linear Au(I) complex, where the gold atom is coordinated with a cyclic (alkyl)(amino)carbene (CAAC) and a carbazolide (Cz) ligands. CMA molecular design results in a diminished overlap integrals between the frontier molecular orbitals ($S_{H/L}$) and achieves fast TADF radiative decay rates up to $4 \cdot 10^6\,\mathrm{s}^{-1}$ with near-unity photoluminescence quantum yields (PLQY)[20]. Moreover, all CMA molecules shows a substantial difference in the orientation of the ground and excited state dipole moment vectors[19], indicating large changes in polarization upon photoexcitation, thus showing a great promise to detect 2PA from classical dipolar CMA materials. This fact triggered numerous attempts in our laboratory to discover a champion 2P-TADF CMA material. We demonstrate the very first dipolar CMA material (LAuCz, Fig. 1d) with enhanced third-order optical nonlinearity, i.e., moderate 2PA cross-section and bright 2P-TADF. Proof-of-concept 2PA response for LAuCz complex was measured in a solid polystyrene matrix showing enhanced 2PA cross-section up to 105 GM in highly diluted thin films (0.1% by weight). Solid state monoliths of the LAuCz complex in polymer shows excellent photostability ($LT_{50}$ = 3 h) vs 20 mW femtosecond pulsed laser excitation at 1000 nm. Our findings establish a new CMA molecular design concept by unlocking bright dipolar 2PA-TADF materials and enabling the development of advanced photonic devices and technologies based on third-order nonlinearity response.

## Results

### Synthesis and structures

Complex LAuCz (Fig. 1d) was prepared from carbazole and the corresponding chloride complex LAuCl in the presence of KO$^t$Bu, following our previously published procedures in high yield[20–23]. Compound LAuCz is an orange solid soluble in toluene and polar aprotic solvents such as dichloromethane (DCM) and tetrahydrofuran (THF). The solid samples are stable in air for months and possess high thermal stability up to 307°C (5% weight loss, Supplementary Fig. S7). Thermogravimetric analysis (TGA) profile for LAuCz samples shows a stepwise decomposition, with the first step accounting for ca. 22% mass loss and attributed to the elimination of the carbazole moiety, based on the 21.6% molecular weight contribution of carbazole. This indicates that the Au–N bond is weaker than the Au–C bond in LAuCz complex. The LAuCz complex possesses $^{13}C_{carbene}$-carbon resonance at 193.3 ppm, which is downfield-shifted compared to analogous CMA complexes based on pyrazine-fused NHC carbene at 186.8–187.6 ppm[24]. This fact suggests less electron shielding for the $C_{carbene}$ atom and stronger π-accepting nature of the π-extended quinoxaline-fused NHC ligand. The single crystal X-ray diffraction shows a linear geometry for complex LAuCz with the Au–N bond length similar to the benchmark complex CMA1 whereas Au–C bond is shorter by 0.05 Å for LAuCz (Supplementary Table S1). This results in 0.07 Å shorter distance between acceptor(carbene) and donor(amide) ligands for complex LAuCz compared to CMA1.

Cyclic voltammetry (CV) was carried out in THF (Supplementary Table S1, Fig. S9). LAuCz shows a quasi-reversible reduction with $E_{1/2}$ at –1.76 V. The energy of the lowest unoccupied molecular orbital (LUMO) is

estimated from the reduction profile onset as –3.72 eV. The reduction process is largely localized over the **L**-carbene ligand which is supported by theoretical calculations (Supplementary Table S1). The LUMO energy of complex LAuCz is stabilised by ca. 1 eV compared to benchmark CMA1 complex (–2.68 eV)[20] reflecting the effect of the π-extended backbone of the carbene with two aza-N atoms acting as a strong-electron withdrawing units. LAuCz exhibits an irreversible oxidation process at +0.36 V which is higher by 0.09 V compared to benchmark CMA1 (Supplementary Table S1). This leads to minor difference for the highest occupied molecular orbital (HOMO) energy (–5.62 eV for LAuCz and –5.53 eV for CMA1, respectively). Complex LAuCz exhibits no change in anodic and cathodic current values during multiple reduction scans (30 scans, Supplementary Fig. S9), indicating excellent electrochemical stability. The HOMO–LUMO energy gap for LAuCz complex of 1.9 eV is nearly 1 eV smaller compared to benchmark CMA1 (2.85 eV) indicating a strongly red-shifted luminescence for the title complex LAuCz.

### One-photon photophysical properties

**Complex LMCl.** We first collected the UV-visible absorption spectrum for the chloride precursor complex LMCl (M = Cu and Au) in various solvents with increasing polarity and performed photophysical characterization (Fig. 2, Table 1, Supplementary Fig. S10). The LCuCl complex shows a sharp absorption band at 350 nm with extinction coefficients (ε) up to $4.0 \times 10^4\,\mathrm{M}^{-1}\,\mathrm{cm}^{-1}$ and a broad low-energy absorption band at ca. 465 nm with ε up to $2.5 \times 10^4\,\mathrm{M}^{-1}\,\mathrm{cm}^{-1}$ (Supplementary Fig. S10). Unlike copper analogue LCuCl, the gold LAuCl complex exhibits a vibronically-resolved absorption band at 350 nm with extinction coefficients (ε) up to $1.5 \times 10^4\,\mathrm{M}^{-1}\,\mathrm{cm}^{-1}$ and weak shoulder at ca. 380 nm. All bands for both complexes demonstrate negligible (up to 5 nm) blue-shift upon increasing the solvent polarity. The natural transition orbitals (NTOs) of the relevant transitions are collected in Fig. 2c and Supplementary Fig. S10. The hole NTO is predominantly localized on metal(I) chloride unit (with 34.3% copper and 21.6% for gold atomic orbitals contribution) while the particle NTO is largely spread across the quinoxaline-carbene ligand. Literature reports for similar carbene-metal(I)-halide complexes[25] ascribe the low-energy band to a halide-metal-to-ligand charge-transfer (XMLCT) transition, while the high-absorption band to an excitation having π–π* LE character because of the higher extinction coefficients for the latter[26].

The photoluminescence profiles for complex LMCl are shown in Fig. 2b (Table 1, Supplementary Fig. S10) for microcrystal powder at room temperature and frozen MeTHF at 77 K. Microcrystals of LMCl (M = Cu and Au) emit orange-red light at 639 nm for copper and 607 nm for gold (weak luminescence) with a broad CT-type emission having monoexponential excited-state lifetime of 3 and 111 μs, respectively. Upon cooling to 77 K, the photoluminescence (PL) profile remains broad while the excited-state lifetime increases up to 13 μs for copper and 18 ms for gold complex. Such minor difference in the PL profile and only several-fold increase in the excited-state lifetime allows us to attribute it to phosphorescence from a $^3$XMLCT state with a weakly involved metal center. All solutions of the LMCl are non-emissive. However, freezing MeTHF solution of LMCl enabled us to detect bright yellow phosphorescence at ca. 500 nm with vibronically resolved profile and very long excited-state lifetime of 32 ms for copper and 5 ms for gold complexes (Fig. 2b, Table 1) which is ascribed to a phosphorescence from the triplet $^3$LE(carbene) state with an energy of 2.54 eV (Table 1). We corroborated this assignment with TD-DFT calculations (Fig. 2c), showing that the NTOs of the triplet excited state are exclusively localized on the carbene ligand (Supplementary Tables S5 and S6).

**CMA complex LAuCz.** UV-vis absorption spectra for LAuCz were measured in solvents with varying polarity: methylcyclohexane (MCH), toluene, tetrahydrofuran (THF) and dichloromethane, as displayed in Fig. 3. Complex LAuCz shows a sharp, intense band at 350 nm which closely resembles that observed in the precursor material LCuCl, showing

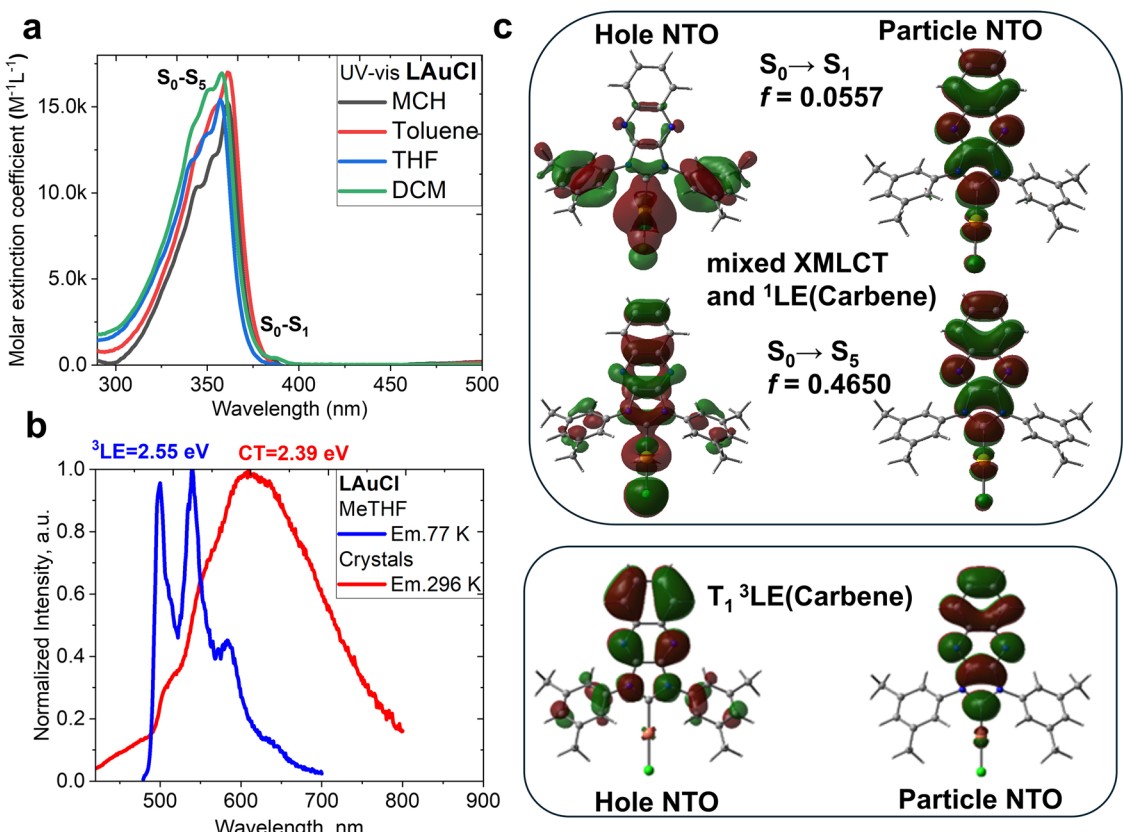

**Fig. 2 | UV-vis, photoluminescence spectra and natural transition orbitals (NTOs) for gold complex LAuCl.** (**a**) UV-vis absorption spectra in various solvents; (**b**) Photoluminescence spectra of **LAuCl** in frozen MeTHF at 77 K and crystals at 296 K (excitation at 365 nm); (**c**) Hole (left) and particle (right) NTO for **LAuCl**.

**Table 1 | Photoluminescent properties of LMCl (M = Cu and Au) and LAuCz in a microcrystalline solid, 0.5% by weight polystyrene matrix (PS), and toluene, THF, DCM solution at 296 and 77 K**

| | $\lambda_{em}$ (nm) | $\tau$ (µs) | $\Phi$ (%)[a] | $k_r$ ($10^6$ s$^{-1}$)[b] | $k_{nr}$ ($10^6$ s$^{-1}$)[c] | $^3$CT/$^1$CT/$^3$LE (eV)[d,e] | $\lambda_{em}$ (nm, 77 K) | $\tau$ (µs, 77 K) |
|---|---|---|---|---|---|---|---|---|
| Crystals, 296 K | | | | | | Frozen MeTHF Glass | | |
| LCuCl | 639 | 3 | 15 | 0.05 | 0.28 | 2.22/–/2.54 | 550 | 32,000 |
| LAuCl | 607 | 111 | <1 | – | – | 2.39/–/2.54 | 500 | 5127 |
| 0.5 wt.% PS matrix, 296 K | | | | | | 0.5 wt. % PS matrix, 77 K | | |
| LAuCz | 613 | 0.354 | 77 | 2.18 | 0.65 | 2.25/2.32/2.53 | 619 | 25 |
| | | | | | | | 507 | 719 |
| LAuCz solution, 296 K | | | | | | Toluene, 77 K | | |
| Toluene | 706 | 0.112 | 19 | 1.70 | 7.23 | 2.40/2.07/2.49 | 593 | 46; 2900 |
| THF | 767 | 0.0066 | 3.7 | 5.61 | 145.9 | 2.36/2.00/2.49 | 602 | 35.3 |
| DCM | 826 | 0.0019 | 0.9 | 4.74 | 521 | 2.20/1.89/– | 627 | 16.9 |

[a]Absolute quantum yields determined using an integrating sphere; [b]Radiative rate constant $k_r = \Phi/\tau$; [c]Nonradiative constant $k_{nr} = (1-\Phi)/\tau$. In case of two-component lifetime $\tau$ an average was used: $\tau_{av} = (B_1/(B_1+B_2))\tau_1 + (B_2/(B_1+B_2))\tau_2$, where $B_1$ and $B_2$ are the relative amplitudes for $\tau_1$ and $\tau_2$, respectively; [d]For LCuCl, energy of $^3$CT and $^3$LE states are obtained from onset values of the emission spectra blue edge in crystals at 296 K and MeTHF glasses at 77 K respectively; [e]For LAuCz, energy of $^1$CT, $^3$CT, and $^3$LE states are based on the onset values of the emission spectra blue edge in 0.5 weight % polystyrene matrix and toluene at 296 ($^1$CT), 77 K ($^3$CT).

negligible solvatochromism with ε up to $2.5 \times 10^4$ M$^{-1}$ cm$^{-1}$, thus attributed to a LE transition mostly involving the carbene unit. The lowest energy band between 400 to 600 nm is broad, unstructured with ε up to $12.5 \times 10^3$ M$^{-1}$ cm$^{-1}$. This low energy band exhibits up to 65 nm blue-shift upon increasing the solvent polarity from methylcyclohexane to dichloromethane (Fig. 3). Such a strong solvatochromism is characteristic for CMA materials[22,23] reflecting the CT nature of this band for complex LAuCz. This empirical observation was corroborated by TD-DFT calculations revealing that this transition has a 1LLCT character and

is dominated (98%) by HOMO to LUMO transition (Fig. 4, Supplementary Tables S5 and S6) with the HOMO and LUMO mostly localized on the carbazole and the L-carbene units, respectively, with only weak contribution from the gold atomic orbitals (Supplementary Table S3). The $^1$LLCT band blue-shift upon increasing solvent polarity is consistent with the large values for the ground (9.9 D) and excited state (13.0 D) dipole moments, with latter having the opposite direction (Supplementary Table S4). Overall, CMA complex LAuCz has strong potential for 2PA due to its large dipole moment changes upon excitation and high ε

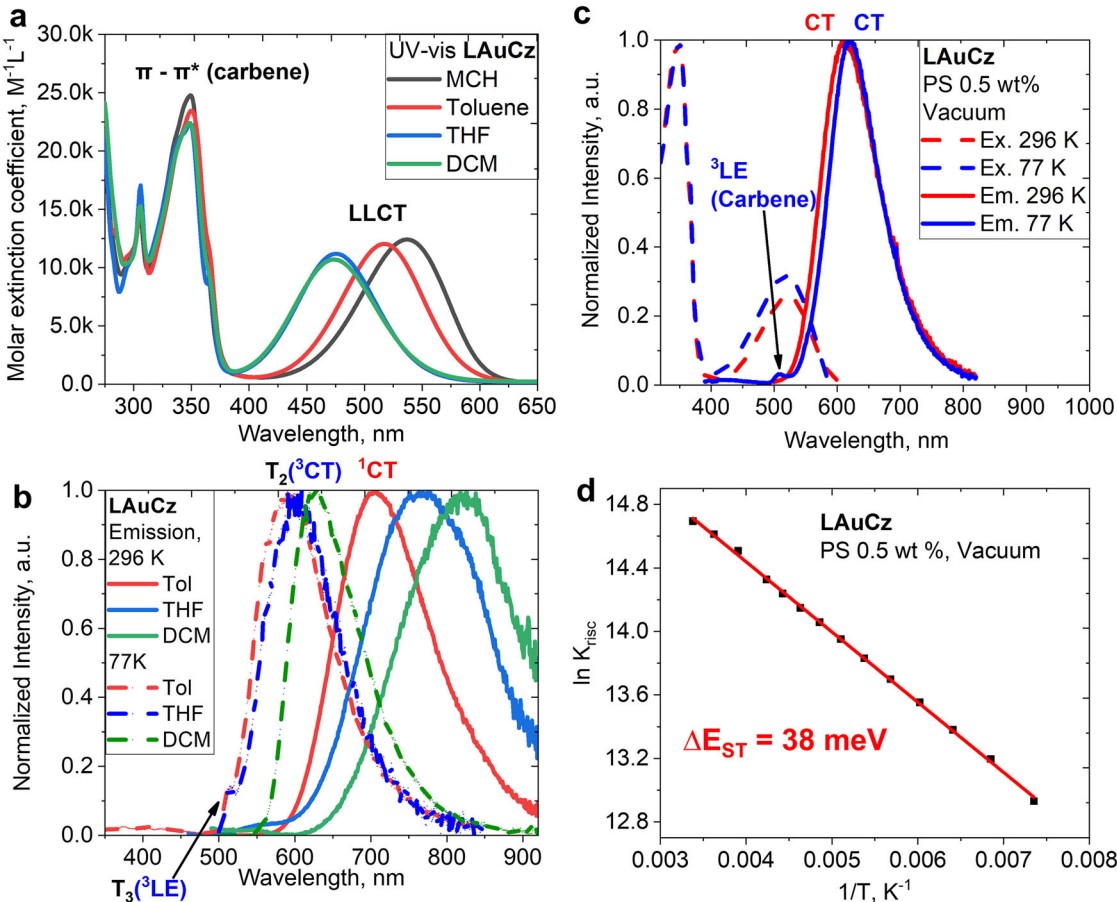

**Fig. 3 | Photophysical properties of LAuCz complex. a** UV-visible absorption spectra for LAuCz in methylcyclohexane (MCH), toluene, tetrahydrofuran (THF), and dichloromethane (DCM); **b** Emission spectra for LAuCz at 296 K (solid lines) and 77 K (dashed lines) in toluene, THF and DCM solution; **c** for 0.5 weight% polystyrene film; **d** Arrhenius fit of temperature-dependent decay time profile to estimate a reverse intersystem crossing (rISC) activation energy ($\Delta E_a$).

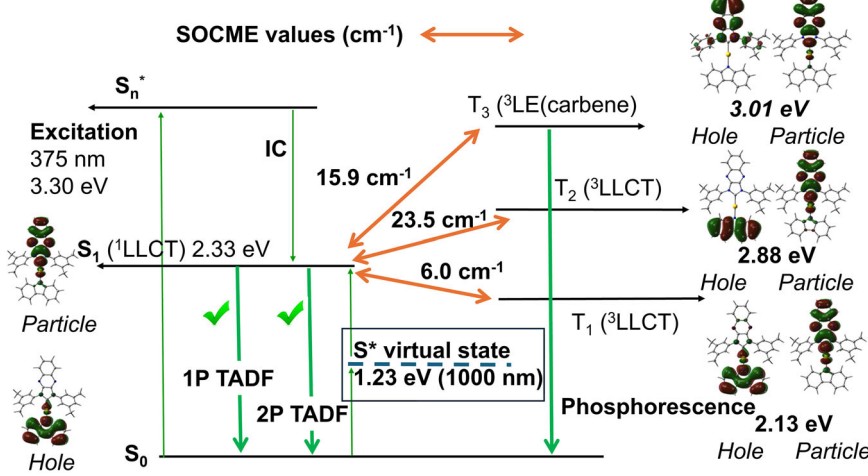

**Fig. 4 | A Jablonski diagram for complex LAuCz.** Excited states energy landscape and NTO (hole and particle) associated with the electronic states $S_1$, $T_1$, $T_2$ and $T_3$ with spin-orbit coupling matrix elements (SOCME, cm$^{-1}$) for complex LAuCz.

up to $12.5\times10^3\,M^{-1}\,cm^{-1}$, which reflect strong polarizability of the CMA molecule comparable with other two-photon absorbing organic dyes ($\varepsilon$ from $10^3$ to $10^5\,M^{-1}\,cm^{-1}$)[27–29].

The photoluminescence of complex LAuCz was measured in a 0.5% by weight polystyrene (PS) matrix and solutions with increasing polarity (toluene, THF and dichloromethane, DCM) at 296 and 77 K with data collected in Table 1 (Fig. 3b, c). The photoluminescence (PL) of LAuCz complex in $1\times10^{-5}$ M toluene solution shows a broad unresolved emission profile at 706 nm with short excited state lifetime of 112 ns and a moderate PLQY of 19% which is four-times lower compared to the PS film (Table 1). This results in a 12-fold increase for non-radiative processes in fluid media ($k_{nr} = 7.2\times10^6\,s^{-1}$) as compared to the solid polystyrene films ($k_{nr} = 0.65\times10^6\,s^{-1}$, Table 1). However, flexible environment for LAuCz molecules in toluene solution enables greater molecular relaxation and reorganizations in the excited state. This further lowers the energy of the $^1$CT state, resulting in 93 nm red-shift into a near-IR region at 767 in THF and 826 nm in DCM

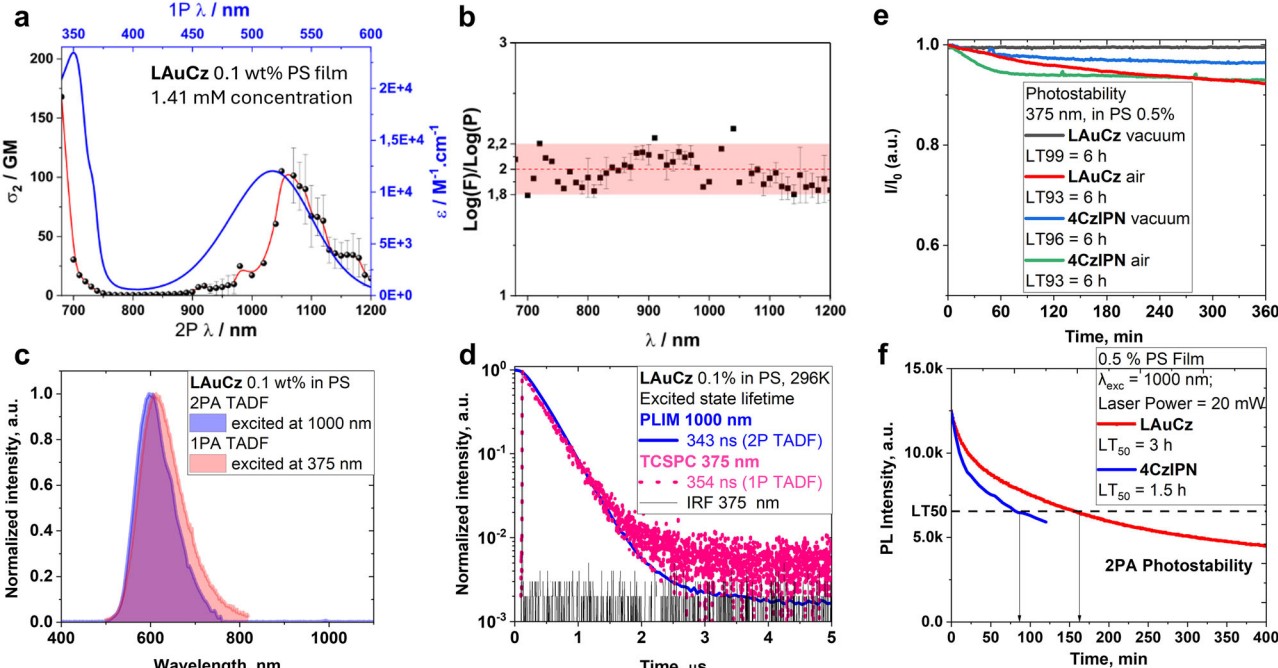

**Fig. 5 | Photophysical properties of LAuCz complex in 0.1% wt. polystyrene (PS) film under argon. a** 1PA vs 2 PA spectra of LAuCz under argon; **b** Quadratic dependence of TADF with excitation power (log(F)/log(P) = 2.0) for LAuCz. 2PA cross-sections and quadratic dependence are displayed as an average of three (2P λ = 790–980 nm)[50] or four (2 P λ = 1070–1200 nm)[51] independent measurements performed on different region in the PS matrix; **c** PL profiles for 1PA-TADF and 2P-TADF experiments excited at 375 nm and 1000 nm, respectively; **d** Excited state lifetime for 1PA TADF from TCSPC experiment and excited at 375 nm and 2P-TADF excited on PLIM with 1000 nm; **e** photostability under constant exposure to the UV-light at 375 nm under vacuum ($5 \times 10^{-5}$ mbar) and air with a comparison to the benchmark 4CzIPN TADF material; **f** exposure to femtosecond laser 20 mW at 1000 nm under argon atmosphere.

solutions of complex LAuCz due to an increasing solvent polarity which additionally stabilizes a highly polar excited state of CMA molecule. Such a large 120 nm red-shift of the emission for LAuCz solutions is paralleled with two-orders of magnitude increase of non-radiative rate from toluene ($k_{nr}$ = $7.2 \times 10^6$ s$^{-1}$) to DCM ($k_{nr}$ = $5.2 \times 10^8$ s$^{-1}$), manifesting the energy gap law correlation. In contrast, polystyrene is a rigid host that restricts conformational changes, favouring brighter and higher-energy emission at 613 nm. To demonstrate the impact of the rigid environment on the photoluminescent properties, we measured frozen solutions PL profiles of LAuCz (Fig. 3b, Table 1), demonstrating up to 200 nm blue-shift and bright luminescence of LAuCz as a broad emission profile at 77 K which becomes similar to the PL profile in a PS-matrix, vide supra. Upon freezing the solutions of LAuCz the emission is dominated by broad phosphorescence from the $^3$CT-state as evidenced by *ca.* 400-fold increase of excited state lifetime up to 46 µs (Table 1) indicating that non-radiative processes have been minimized.

At 296 K in PS matrix, complex LAuCz shows bright red emission at 613 nm with a high PLQY of 77% and short excited-state lifetime of 354 ns. The PL profile is broad and featureless, indicating a CT emission with one of the highest radiative rates reported across the CMA materials, $k_r$ = $2.1 \cdot 10^6$ s$^{-1}$ (Table 1). Upon cooling to 77 K, LAuCz exhibits a 6 nm blue-shift for the featureless CT profile and low intensity shorter wavelength emission at 507 nm. The latter exhibits an emission lifetime of 719 µs, with energy 2.53 eV determined as an onset of the blue emission profile edge. Note the precursor chloride complex LAuCl shows similar energy of 2.54 eV in frozen-MeTHF, vide supra. Therefore, we assigned the high energy emission at 507 nm to phosphorescence from the $^3$LE(Carbene) state. The CT emission at 619 nm for complex LAuCz at 77 K experiences 70-fold excited-state lifetime increase up to 25 µs from 354 ns at room temperature. Such a strong elongation of the excited-state lifetime is characteristic for the TADF CMA materials (Fig. 5, Supplementary Fig. S11)[24]. Therefore, we assign the broad, low-energy emission to TADF from a singlet $^1$CT state at 2.32 eV. Oxygen quenching experiments has been carried out for LAuCz samples to

further confirm the involvement of the triplet states (Supplementary Fig. S25) by demonstrating the total quenching of the phosphorescence from $^3$LE(Carbene) and reduced intensity of TADF from CT-states.

The luminescent properties for LAuCz in PS matrix were measured as a function of temperature to determine the activation energy barrier of the reverse intersystem crossing (rISC) process for TADF. The excited state lifetime of LAuCz markedly increases after *ca.* 100 K (Supplementary Fig. S11), albeit emission remains as a mix of TADF and phosphorescence. The emission becomes predominantly phosphorescent on cooling down to 16 K as reflected by a dramatic increase in the excited state lifetime up to 147 µs. A plateau corresponding to pure phosphorescence has not been reached at 16 K, suggesting that TADF remains present even at such a low temperature. We directly fit the temperature-dependent PL decay curves to the Arrhenius equation of ln $k_{RISC}$ vs 1/T (Fig. 3d) based on a two-level S$_1$/T$_1$ model[30] to estimate the singlet-triplet gap ($\Delta E_{ST}$) of LAuCz as 38 meV. The benchmark CMA1 material exhibits a two-times larger $\Delta E_{ST}$ value of 69 meV[20], therefore suggesting that LAuCz has faster rISC rates. We explain such excellent performance for complex LAuCz based on our computational results indicating a smaller overlap integral for LAuCz ($S_{H/L}$ = 0.26) compared to benchmark CMA1 complex ($S_{H/L}$ = 0.36). Small $\Delta E_{ST}$ and $S_{H/L}$ are important but not sufficient criteria to unlock CMA TADF materials with fast radiative rates exceeding $10^6$ s$^{-1}$. We demonstrated based on various Aza-CMA materials that the negative energy gap between CT and 3LE states, $\Delta E(CT–3LE)$ is a second key parameter to check. The larger the $\Delta E(CT–3LE)$ the faster radiative rates for the CMA emitters. Complex LAuCz has $\Delta E(CT–3LE)$ equal to −0.2 eV similar to benchmark CMA1 material explaining high radiative rate of $2.18 \times 10^6$ s$^{-1}$. Notably, the radiative rate surpasses the non-radiative rate, $k_{nr}$ = $0.65 \times 10^6$ s$^{-1}$, suggesting a high rigidity of the geometry of LAuCz molecule in the PS matrix once in the excited state.

Transient PL decay analysis enabled estimation of the prompt fluorescence (PF) excited state lifetime of *ca.* 600 ps which is four-orders of magnitude faster compared to delayed fluorescence (DF) of 354 ns

(Fig. S24). The key TADF rate constants (Table S11) were calculated using a well-established model[31]. The $k_{rISC}$ value was estimated as $1.67 \times 10^9\,s^{-1}$, which is close similar with other CMA emitters ($0.8–3.5 \times 10^9\,s^{-1}$)[32] and orders of magnitude faster compared to classical organic TADF and 2 P TADF molecules ($k_{rISC} = 10^4–10^6\,s^{-1}$). High $k_{rISC}$ of LAuCz is attributed to near thermal equilibrium between singlet and triplet LLCT excited states and strong spin-orbit coupling of the bridging gold(I) atom. Such a fast $k_{rISC}$ rate translates into superior photostability of the LAuCz (0.1 wt.% doped PS film, Fig. 5e) compared to benchmark organic TADF emitter 4CzIPN. Under constant exposure to 375 nm UV light, the PS films of the title complex show almost no photodegradation under vacuum with only small photodegradation in air (LT$_{93}$ of 6 h, where LT$_{93}$ is time taken to reduce initial brightness by 7%). These results demonstrate that higher $k_{rISC}$ rate of the organometallic TADF compared to organic TADF emitters effectively reduces photodegradation rate and enhances TADF properties in thin film[19]. This superior photostability is also beneficial for improving the 2 P TADF photostability, vide infra, which is crucial for future practical applications.

Excited-state TD-DFT calculations were performed to investigate the energetic ordering and nature (CT or LE) of the excited states of the LAuCz complex (Fig. 4, Supplementary Tables S5 and S6). The reference material CMA1 shows first singlet (S$_1$) and triplet (T$_1$) excited states with CT character, while the second triplet (T$_2$) excited state is commonly locally excited amide ligand $^3$LE(amide)[20,23]. Up to now, all complexes followed such ordering. On the contrary, the first two triplet states (T$_1$ and T$_2$) of LAuCz exhibit a CT character originating from HOMO → LUMO and HOMO−1 → LUMO transitions, respectively, while T$_3$ corresponds to the $^3$LE state located on the π-extended quinoxaline-carbene ligand. The calculated spin-orbit coupling matrix elements (SOCME, cm$^{-1}$, Supplementary Table S7) for complex LAuCz are consistently high between singlet and higher energy triplet (T$_2$ and T$_3$) states up to 31.3 cm$^{-1}$ (Fig. 4). Even though T$_2$ and T$_3$ are much higher in energy as compared to T$_1$ making the direct population of these states unlikely, the large SOCME between S$_1$ and T$_{2,3}$ together with a large vibronic coupling between T$_1$ and T$_{2,3}$ could further enhance rISC rate between T$_1$ and S$_1$ in the LAuCz material[33].

**Transient absorption and two-photon absorption**. We measure 2PA for LAuCz in various media encouraged by its superior radiative rates up to $2.1 \times 10^6\,s^{-1}$, high polarizability and extinction coefficients which are proportional to the square of the transition dipole moment ($|\mu|^2$) for S$_0$ → S$_1$ transition[26] – as one of the conditions for large 2PA cross-sections ($\sigma_2$). Thanks to bright TADF from LAuCz, the 2PA spectra were measured using 2PEF method in degassed solution (toluene, THF) and polystyrene matrix (PS, 0.1% weight of LAuCz), the results are presented on Fig. 5. Confirmation of simultaneous two-photon absorption is evidenced by a quadratic dependence between upconverted fluorescence intensity Log(F) and excitation laser light intensity Log(P) for compound LAuCz (Fig. 5b, Supplementary Figs. S13–15 for all measured wavelengths from 680 to 1200 nm). The 2PEF of LAuCz perfectly overlaps with the 1PA emission (Fig. 5c), corroborating TADF nature for the 2PEF signal. Although a low $\sigma_2^{max}$ value was measured in solution (ca. $\sigma_2^{max} < 0.5$ GM, Fig. S11), the 2PA response of LAuCz is significantly enhanced in a PS matrix resulting in a moderate 105 GM at 1050 nm-corresponding to an increase of about three orders of magnitudes. Enhanced 2PA cross-section value parallels well with the high radiative rates for complex LAuCz in rigid polystyrene matrix that maintains a planar or near-planar geometry. Favourable molecular alignment or aggregation effects lead to significantly enhanced 2PA cross-sections[30]. To rule out aggregation enhancement of the TPA TADF for LAuCz, grazing incidence wide-angle X-ray scattering (GIWAXS, Supplementary Fig. S23) measurements were performed. The scattering pattern of the LAuCz in host PS films does not possess any significant ordering that would be commensurate with distinct, LAuCz aggregates and that there is no discernible ordering or orientation of the LAuCz in the PS host. As a control a neat LAuCz film possess significant crystalline features

commensurate with distinct crystalline, ordering of LAuCz. As such features are not observed in the LAuCz:PS blend these GIWAXS measurements enable us to rule out aggregation enhancement in this system.

2PA cross-section calculations were carried out at the TD-DFT level of theory in gas-phase and toluene (see the methods section for further details) for LAuCz (Supplementary Fig. S18–22 and Supplementary Table S8). The simulated 2PA spectrum in toluene shows a low-energy band centred at ~800 nm (247 GM) composed solely of the S$_0$-S$_1$ excitation, described mainly by a HOMO-LUMO transition. The absence of surrounding polarization in gas-phase decreases the transition energy, the dipole moment, and the transition dipole moment of the LAuCz, which are characteristic quantities for charge-transfer excitations and directly connected to the 2PA transition moments. Therefore, the 2PA spectrum in gas-phase results in a red shift of about 100 nm and a decrease of 25% in the intensity, indicating LAuCz 2PA properties are strongly sensitive to the environment. Besides, the nature of the excitation is not affected by the solvent and remains dominated by a HOMO-LUMO transition. The second and higher-energy band (~ 450–650 nm) comprises several transitions challenging a detailed characterization. Moreover, the incident photon energy approaches the S$_0$–S$_1$ excitation energy, and 1PA resonant effects are expected to contribute to the intensity of the band. The few-states model was applied in order to unravel the dominant and possible interference channels involved in the S$_0$–S$_1$ 2 PA transition (Supplementary Tables S9 and S10). The two-state model reveals a large S$_1$ contribution both in toluene (426 GM) and in gas-phase (260 GM) compared to the values obtained using the time-dependent quadratic response formalism. These results are expected due to the observed strong 1-dimensional CT character of the S$_0$–S$_1$ transition, which leads to large transition dipole moments and large variation of the permanent dipole moments in the same direction. Then, considering the five lowest-lying excited states (S$_{0-5}$), the three-states model shows that the S$_5$ state is a strong interference channel in toluene, leading to a 2PA cross-section of 333 GM (including S$_0$, S$_1$, and S$_5$) and a slight increase of 5 GM for the isolated LAuCz. Finally, including the 5 lowest-lying excited states (a six-state model including S$_0$-S$_5$) leads to a value of 330 GM (258 GM), which are larger by a factor of 1.33 (1.39) with respect to the reference value of 247 GM (183 GM) in toluene (isolated) obtained using the time-dependent quadratic response formalism. These results indicate large contributions from high-energetic interference channels beyond the S$_5$ state and from the surrounding effects. Moreover, these analyses were carried out on a static optimized geometry representing the ground state, and the dynamic effects due to conformational sampling were not addressed[34]. Therefore, a lower 2PA cross-section amplitude is expected when considering the dynamic nature of the system in solution[35] which would rationalize the difference between the theoretical and the experimental estimates. Furthermore, a direct comparison with the solid-state results is challenging since it would require the adaptation of Eqs. (1 and 3, see Methods) to account for the presence of an anisotropic medium in addition to the simulation of the emitting layer morphology, which are beyond the aim of this study.

Transient absorption (TA) spectra for complex LAuCz in the PS matrix at 298 K are plotted in Fig. 6. The spectra on picosecond time scales show the initial excited state absorption at 600–1250 nm within 1 ps. The initial signal observed in TA in the 600 to 800 nm range is predicted from TD-DFT calculations (see methods section for computational details) to arise from the S$_1$ excited-state absorption. The band centred at 650 nm is associated with triplet states and appears after 100 ps. Calculations revealed that this band could originate from the absorption of either the T$_1$ or T$_2$ states while the T$_3$ state has a very weak absorption and might not be observed in TA-experiment. Moreover, we measured 2PA emission-excitation map indicating that the highest 2PA efficiency can be measured below 680 nm followed by the range from 900 to 1200 nm (Fig. 6a) thus in excellent agreement with the singlet and triplet excited state absorptions (Fig. 6b).

To reveal the nature of the 2PA emission we first overlaid 1PA profile excited at 375 nm with the 2PA PL profile (850 nm excitation, Fig. 5c). Both PL profiles perfectly overlap demonstrating a broad CT emission. Next, we

   

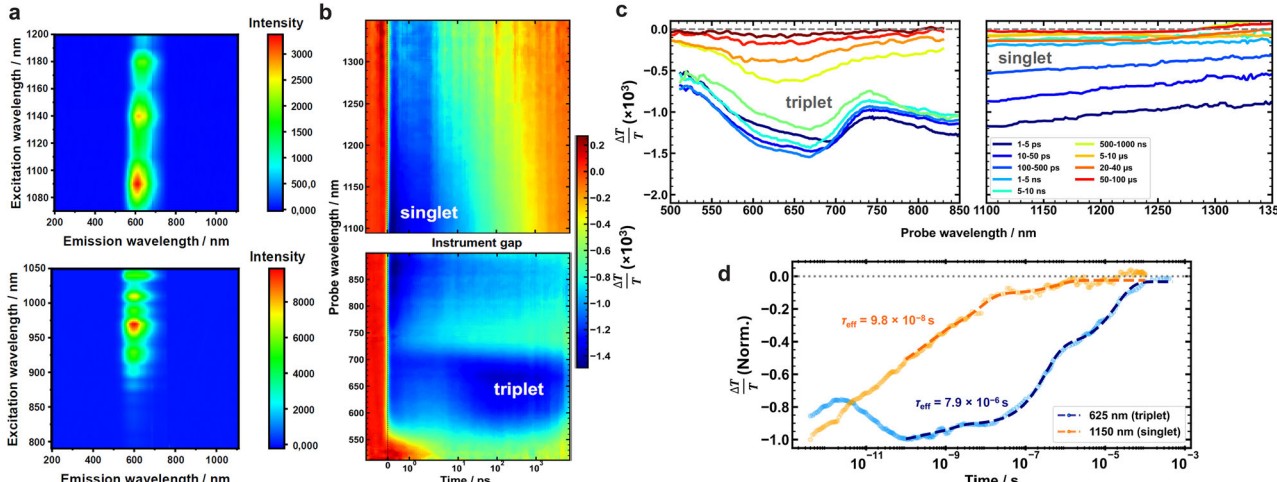

**Fig. 6 | 2PA and transient absorption (TA) data for complex LAuCz in a 0.1% wt. polystyrene (PS) film. a** Log-scaled LAuCz 2PA emission-excitation map; emissions were corrected by input power and emission over 650 nm, where excitation range 680–1050 nm is cropped by short pass optical filter, **b** Transient absorption spectroscopy map of LAuCz film (0.1 wt% % load in PS) at room temperature, following 370 nm excitation at a fluence of 9.4 μJ cm$^{-2}$. PIA bands corresponding to singlet and triplet excitons are centred at 1150 and 625 nm, respectively; **c** Time slices of normalised spectra. **d** Kinetic traces of singlet and triplet PIAs extracted from picosecond-microsecond TA measurements with corresponding bi- and triexponential fits. After rapid intersystem-crossing, the triplet population peaks around 100 ps.

measured 2PA excited state lifetime from complex LAuCz in PS film on phosphorescence lifetime imaging setup (PLIM, Fig. 5c). Monoexponential fitting and overlay of the traces from the 1PA experiments excited at 375 nm and 1000 nm femtosecond pulsed laser results in nearly identical 1PA (354 ns) and 2PA (343 ns) excited decay time. This decay timescale is attributed to the delayed fluorescence arising from the upconversion of the dark triplet into emissive singlet considering that the prompt fluorescence occurs on a much faster timescale from complex LAuCz for both 1PA and 2PA events. Films of LAuCz in PS were irradiated by a 1000 nm femtosecond pulsed laser to measure excellent photostability of 3 h (LT$_{50}$) which opens future 2 PA application opportunities in advanced photonic technologies relying on stable materials with strong third-order optical nonlinearity.

## Discussion

We present a molecular design strategy for a thermally and photostable dipolar gold CMA complex, representing the first CMA material to exhibit third-order nonlinear optical properties with bright red two-photon–excited TADF (2P-TADF). It was found that benchmark CMA1 complex, despite possessing a large transition dipole moment ($|\mu|^2$ for S$_0 \rightarrow$ S$_1$) and high TADF radiative rates, fails to emit 2P-TADF, pinpointing that the ambiphilicity of the carbene ligand alone is insufficient. It's the polarizability that must be taken into account; therefore, the use of simultaneously strong ambiphilic and π-extended carbene moieties, for instance, with a quinoxaline backbone, enabled the first proof-of-concept demonstration of highly sought-after solid-state CMA two-photon absorption (2PA) and 2P TADF material.

This success was achieved by doping LAuCz complex in the PS matrix, exhibiting an enhanced 2PA cross-section up to 105 GM and photostability (LT$_{50}$ = 3 h) towards 20 mW femtosecond pulsed laser excitation at 1000 nm, which is two time longer compared to benchmark 4CzIPN organic TADF material. It was found that $k_{rISC}$ of LAuCz complex (1.67 × 10$^9$ s$^{-1}$) is orders of magnitude higher compared to classical organic TADF materials thanks to the bridging gold atom with high spin-orbit coupling coefficient. We correlate high $k_{rISC}$ rate for title molecule with a ten-fold higher photostability compared to classical dipolar organic TADF molecules, thanks to faster harvesting of the triplet states. Moreover, 2PA cross-section extends into the near-infrared II region up to 1300 nm with a significant CT character that enhances the 2PA process, inviting multiphoton

absorption experiments in the future. Unlike benchmark material CMA1, complex LAuCz features a bright carbene-centred $^3$LE state which is 0.2 eV above the CT state, while showing a very small $\Delta E_{ST}$ of 38 meV and ensuring fast rISC rate for bright red TADF emission in thin films with PLQY up to 77%.

We demonstrated that complex LAuCz emits bright red TADF and 2 P TADF as a result of the one and two-photon excitation with exceptional radiative rates of up to 2.1·10$^6$ s$^{-1}$, making LAuCz one of the best deep red CMA TADF emitters. The singlet CT nature of the 2 PA excitation channel has been confirmed by quantum-chemical calculations and correlated with near-infrared TA measurements and 2PA emission-excitation map. This work establishes molecular design guidelines for developing future bright dipolar 2PA and 2P TADF organometallic chromophores, tailored for advanced photonic applications.

## Methods
### General
**Computational details.** The excited-state S$_1$ and T$_1$ structures of the LAuCz complex were optimized at the time-dependent density functional theory (TD-DFT) level by using the Tamm-Dancoff approximation (TDA), the exchange-correlation ωB97X-D functional, and the 6-31 G(d,p) basis set for all the atomic species, except Au for which the def2-TZVP basis set was used along with effective core potential. Excited-state properties were computed at the optimally tuned screening range-separated hybrid LC-ωhPBE level of theory in toluene, by using the 6-311 G(d,p) basis set for all the atomic species, except for Au (as stated above). For further details we refer to our previous work[36]. The Multiwfn[37] tool was then used to post-process the Gaussian16[38] output and to extract excitation energies, oscillator strengths, and transition dipole moments between excited states. Hole-electron density overlaps were estimated by the Φ$_s$ metric, according to the detachment and attachment formalism described elsewhere[39]. The Supplementary Data 1 contains cartesian coordinates of the optimized structures.

The 2PA cross-section spectra were simulated for the optimized geometry obtained in gas phase. All the 2PA calculations were carried out in Dalton2020.1[40] software using the CAM-B3LYP DFT functional and the 6–311 G(d,p)//def2-TZVP basis sets. The choice of the CAM-B3LYP was based on the requirement of including long-range electronic interactions and was based on direct theoretical-experimental comparisons[41–43] and by

**Article**

benchmarks against high-level Coupled Cluster computations[44–46]. 2PA transition probabilities of the first 20 excited states were calculated using the single residue of the time-dependent quadratic response vectors[47]. In addition, for the few-states models, the permanent dipole moment of the excited states and the transition dipole moments between excited states up to the fifth lowest-lying transitions were computed using the double residue scheme. Solvent effects were taken into account using the continuum FixSol[48] model implemented in the Polarizable Embedding library[49]. Finally, the 2PA cross-sections were computed assuming a single beam experiment[44] as given by

$$\sigma_{2PA}^{ng}(\omega) = \frac{4\pi^3 \omega_{ng}^2}{c^2} \, \delta_{2PA}^{ng} \, g(2\omega, \omega_{ng}; \Gamma) \quad (1)$$

$$g\left(2x, \omega_{ng}; \Gamma\right) = \frac{\sqrt{\ln 2}}{\sqrt{\pi}} \frac{2}{\Gamma} \, e^{-\frac{\ln 2 \left(2x - \omega_{ng}\right)^2}{(\Gamma/2)^2}} \quad (2)$$

$$\delta_{2PA}^{ng} = \frac{1}{15} \sum_{i,j=x,y,z} \left\{ S_{ii}^{ng} \left(S_{jj}^{ng}\right)^* + 2 S_{ij}^{nh} \left(S_{ij}^{ng}\right)^* \right\} \quad (3)$$

$$S_{ij}^{ng} = \sum_k \left\{ \frac{\mu_{gk}^i \mu_{kn}^j}{\omega_{ng} - \omega} + \frac{\mu_{gk}^j \mu_{kn}^i}{\omega_{ng} - \omega} \right\} \quad (4)$$

where $\sigma_{2PA}^{ng}(\omega)$ is the spectral contribution from the $g \to n$ transition with $\hbar\omega_{ng}$ transition energy and $\delta_{2PA}^{ng}$ transition probability. $c$ is the speed of light in vacuum, $g(2\omega, \omega_{ng}; \Gamma)$ is a broadening function with a full-width at half-maximum parameter ($\Gamma = 0.3$ eV in this study), $S_{ij}^{ng}$ is the two-photon transition moment, and $\mu_{ab}^i$ is the $i$ component of the transition dipole moment between the $a$, $b$ states or permanent dipole moment when $a = b$.

## Data availability

The data that support the plots within this paper and the Supplementary Information and other findings of this study are available from the corresponding authors upon reasonable request.

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

## Acknowledgements
A.S.R. acknowledges support from the Royal Society (grant nos. URF\R1\180288, RGF\EA\181008, URF\R\231014), EPSRC (grant code EP/K039547/1 and APP46952). M.L. acknowledges the Academy of Finland Flagship Programme, Photonics Research and Innovation (PREIN), decision 320166, the Finnish Grid and Cloud Infrastructure resources (urn:nbn:fi:research-infras-2016072533). N.L.P. acknowledges the Doctoral Programme in Science, Forestry and Technology (Lumeto, University of Eastern Finland). T.N.R. is a postdoctoral researcher of the Fonds de la Recherche Scientifique – FNRS" (F.R.S.–FNRS). We thank Dr Louise Natrajan, EPSRC and University of Manchester for access the Centre for Radiochemistry Research National Nuclear User's Facility (NNUF, EP/T011289/1) to use FLS-1000 fluorometer. D.T.W.T. acknowledges Diamond Light Source for access to the DL-SAXS equipment (experiment number SM40538-1) supported by an EPSRC grant (EP/R042683/1), and instrument scientist Dr Paul Wady for their help and support during beamtime. L.M. thanks the Winton Programme and Harding Distinguished Postgraduate Scholarship for funding. A.J.G. thanks the Leverhulme Trust for an Early Career Fellowship (ECF-2022-445), the Knut and Alice Wallenberg Foundation for a Wallenberg Academy Fellows award (KAW 2023.0082), and the Swedish Research Council (VR) for a Starting Grant (2024-03915). European Union's Horizon 2020 research and innovation programme grant agreement no. 101020167 (L.M. and A.J.G.).

## Author contributions
I.D.N. carried out synthesis and characterization, electrochemistry, UV-vis and photoluminescence spectroscopy and analysis, L.M. and A.J.G. performed transient absorption measurements and analysis, D.T.W.T. performed GIWAXS measurements and analysis, T.N.R., N.L.P., G.L., M.L., Y.O. performed theoretical calculations for the ground and excited states and calculations involving a two-photon process, C.T.S. assisted with initial collection of the two-photon absorption data and performed two-photon excited TADF lifetime measurements, G.F.S.W. performed single crystal experiment and refinement, M.B.-D., J.D. and I.D.N. performed two-photon absorption and luminescence experiments in fluid and solid state, A.S.R. conceived and designed the idea. J.D., M.L., Y.O. and A.S.R. planned the project and designed the experiments. I.D.N., Y.O., and A.S.R. wrote the manuscript. All authors contributed to the discussion of the results, analysis of the data, and reviewed and corrected the manuscript.

## Competing interests
The authors declare no competing interests.
