## [Transparent Peer Review file · Communications Chemistry]

Enhanced Third-Order Optical Nonlinearity in a Dipolar Carbene-Metal-Amide Material with Two-Photon Excited Delayed Fluorescence

Corresponding Author: Dr Alexander Romanov

This manuscript has been previously reviewed at another journal. This document only contains information relating to versions considered at Communications Chemistry.

Version 0:

Reviewer comments:

Reviewer #2

(Remarks to the Author)

The overall scope and direction of the manuscript align with the aims of Communications Chemistry, and the authors have addressed several of the reviewers' comments concerns to some extent. However, a number of important technical points still require clarification or further validation.

1. Photodegradation experiment (Fig. 5e):

Please specify the light intensity used during the photodegradation measurements. In addition, could the authors clarify the expected photodegradation mechanism? A brief discussion or supporting evidence would strengthen the interpretation.

2. Comparison of delayed fluorescence under 1PA and 2PA (Fig. 5d):

The authors are requested to confirm whether the delayed component is quenched in the presence of oxygen. This verification is necessary to ensure that the long-lived emission originates from genuine delayed fluorescence, rather than artifacts such as light scattering within the solid film.

3. Thermogravimetric Analysis (SI, line 139):

The TGA curve appears to show multiple weight-loss steps. What decomposition products are anticipated in each step? Furthermore, how do the observed decomposition temperatures compare with those of structurally related derivatives?

4. Electrochemical stability (Fig. S9):

The authors describe the material as having "high electrochemical stability", but the initial reduction peak appears irreversible, suggesting possible decomposition. Please clarify how this observation is reconciled with the stated stability.

Version 1:

Reviewer comments:

Reviewer #2

(Remarks to the Author)

The authors adequately addressed my previous comments.

As a minor technical point, I suggest including the instrument response function for the lifetime measurements in Figure 5d.

Reviewer #2 (Remarks to the Author):

The overall scope and direction of the manuscript align with the aims of Communications Chemistry, and the authors have addressed several of the reviewers' comments concerns to some extent. However, a number of important technical points still require clarification or further validation.

We thank reviewer for the comments aiming to improving the manuscript. Please find below our answers to the technical comments.

1. Photodegradation experiment (Fig. 5e): Please specify the light intensity used during the photodegradation measurements. In addition, could the authors clarify the expected photodegradation mechanism? A brief discussion or supporting evidence would strengthen the interpretation.

Response: Agreed and added description. The light intensity was measured at ca. 46 $\mu\text{W}/\text{cm}^2$ with the ThorLabs S120VC standard photodiode power sensor.

Added text in supporting information: "Steady-state PL spectra were recorded using an Edinburg Instruments FLS1000 spectrofluorimeter. The light source was a monochromated 450 W Xenon arc lamp; excitation wavelength varied. The photostability experiments have been carried out with the excitation slit set to 1 for a 375 nm wavelength. The light intensity was measured at ca. 46 $\mu\text{W}/\text{cm}^2$ with the ThorLabs S120VC standard photodiode power sensor. Samples were measured in air, vacuum or under flowing nitrogen at room temperature."

The mechanism of photodegradation falls outside the scope of this research manuscript, which has been focused on demonstrating the molecular design that results in the first bright and robust 2PA TADF CMA material. However, we have evidence for the first degradation step from the TGA experiment, indicating that decomposition is likely initiated by the elimination of the carbazole moiety (see below).

2. Comparison of delayed fluorescence under 1PA and 2PA (Fig. 5d):

The authors are requested to confirm whether the delayed component is quenched in the presence of oxygen. This verification is necessary to ensure that the long-lived emission originates from genuine delayed fluorescence, rather than artifacts such as light scattering within the solid film.

Response: Agreed and added data oxygen quenching experiment in supporting information Figure S25. This comment actually raises a major fundamental difference between organic and organometallic TADF materials and difference in their response to oxygen quenching experiment, which is worth a tutorial paper on TADF material photophysics.

The effect of oxygen quenching is frequently used to confirm, and quantify, the contribution of delayed fluorescence (DF) to the overall emission of organic TADF materials. This is due to TADF and phosphorescence both involve the triplet state, the presence of oxygen tends to quench the TADF and phosphorescence emissions (Advanced Science, 2016, 3, 1600080). However, as oxygen tends to quench long-lived emissions in general, in some instances, especially in a solution, PF quenching also occurs (Journal of Physical Chemistry C, 2018, 122, 23934–23942; Angewandte

Chemie International Edition, 2023, 62, e202217530). Therefore, this experiment should be used with extreme caution and only as a rough indicator, but not as a proof of triplet emission on its own due to immensely small prompt component compared for delayed component in organometallic TADF materials.

Moreover the extent of oxygen quenching is significantly less pronounced for TADF emitters with submicrosecond lifetime and fast radiative rates exceeding 10^6 s^{-1} . For instance, we previously performed such oxygen quenching for a range of CMA emitters with a varied excited state lifetime (0.9 to tens of microsecond excited state lifetime, Adv. Optical Mater. 2022, 10, 2200312). In this work we demonstrated that “fast” CMA emitters result only in small intensity drop of the broad CT profile, whereas “slow” CMA emitters may reveal a significant drop in emission intensity and reveal vibronically resolved triplet state $^3\text{LE}(\text{amide})$.

We repeated the oxygen quenching experiments is demonstrated in Figure S25 panels b and c. Added panels b and c:

Added text in manuscript: “Oxygen quenching experiments has been carried out for **LAuCz** samples to further confirm the involvement of the triplet states (Figure S25) by demonstrating the total quenching of the phosphorescence from $^3\text{LE}(\text{Carbene})$ and reduced intensity of TADF from CT-states.”

Therefore, methods based on time-resolved and varied temperature luminescence are thus preferable. To support the TADF mechanism assignment we demonstrated a classical sigmoidal profile for the excited state lifetime vs temperature (Figure S11) which is classical for CMA TADF materials in line with numerous reports from our team and others. We also applied a combination of the varied temperature and time-resolved spectroscopy to successfully analyse TADF kinetics (Figure S24, S25; C. Adachi, *J. Phys. Chem. A* **2021**, 125, 36, 8074-8089), which was requested during previous revisions. All experimental facts confirm the TADF nature of the luminescence for the title complex **LAuCz**, ruling out scattering interpretation.

3. Thermogravimetric Analysis (SI, line 139):

The TGA curve appears to show multiple weight-loss steps. What decomposition products are anticipated in each step? Furthermore, how do the observed decomposition temperatures compare with those of structurally related derivatives?

Response: The decomposition temperatures of the title **LAuCz** complex (307°C) is on the side of the most stable CMA emitters reported by our team and others, varying on average from 180 to 340°C. The stepwise decomposition has been analysed where the first step can be attributed to the loss of the carbazole-moiety based on ca. 22% mass loss which is exactly corresponds to carbazole moiety. It's not obvious what entity may correspond to 18% mass loss on the second step as it may well be associated with fragmentation of the remaining carbene moiety. Therefore, we only add description for the first step.

Added text:

“Thermogravimetric analysis (TGA) profile for **LAuCz** samples shows a stepwise decomposition, with the first step accounting for ca. 22% mass loss and attributed to the elimination of the carbazole moiety, based on the 21.6% molecular weight contribution of carbazole. This indicates that the Au–N bond is weaker than the Au–C bond in **LAuCz** complex.”

Figure S7 and caption has been updated

Figure S7. TGA curve for **LAuCz**, showing a stepwise decomposition. Decomposition temperature (T_d) indicates the temperature at 5% weight loss. The first decomposition step accounts for ca. 22% mass loss which can be attributed to the elimination of carbazole-moiety based on 21.6% molecular weight contribution.

4. Electrochemical stability (Fig. S9):

The authors describe the material as having “high electrochemical stability”, but the initial reduction peak appears irreversible, suggesting possible decomposition. Please clarify how this observation is reconciled with the stated stability.

Response: It's likely there might be a confusion between reduction (quasi-reversible process) and oxidation (irreversible) processes. The reduction peak is quasi-reversible as demonstrated in Figure S9 “The near-unity ratio between cathodic and anodic currents i_{pc}/i_{pa} of 0.71 supports the quasi-reversible character of the reduction process (left) and remains unchanged after 30 reduction cycles indicating high electrochemical stability of the complex **LAuCz** (right).”

We refer only to reduction process in the main text of the article: “Complex **LAuCz** exhibits no change in anodic and cathodic current values during multiple reduction scans (30 scans, Figure S9), indicating excellent electrochemical stability.”

Reviewer #2 (Remarks to the Author):

The authors adequately addressed my previous comments. As a minor technical point, I suggest including the instrument response function for the lifetime measurements in Figure 5d.

Response: We thank reviewer for all comments and time devoted to enhance the quality of the manuscript. Agreed and added instrument response function for the lifetime measurements, please see updated Figure 5d below.

EDITORIAL REQUESTS

In addition, please ensure that the remaining instances where the 2PA cross-section is referred to as "high" (in lines 126, 341 and 441) are revised (one of the reviewers had previously recommended to use "moderate" instead).

Response: We agree and corrected remained instances of "high" to either "moderate" or "enhanced" where it was more appropriate.

At the same time we ask that you edit your manuscript to comply with our journal policies and formatting style in order to maximise the accessibility and therefore the impact of your work.

Response: We agree and provide separate word file with a check list for all editorial requests with this revision.